# Quality Evaluation of *Gastrodia Elata* Tubers Based on HPLC Fingerprint Analyses and Quantitative Analysis of Multi-Components by Single Marker

**DOI:** 10.3390/molecules24081521

**Published:** 2019-04-17

**Authors:** Yehong Li, Yiming Zhang, Zejun Zhang, Yupiao Hu, Xiuming Cui, Yin Xiong

**Affiliations:** 1Faculty of Life Science and Technology, Kunming University of Science and Technology, Kunming 650500, China; jr93586@163.com (Y.Z.); 18380802826@126.com (Z.Z.); hypflygo@163.com (Y.H.); cuisanqi37@163.com (X.C.); 2School of Pharmacy, China Pharmaceutical University, Nanjing 210009, China; yhlcpu@163.com; 3Yunnan Key Laboratory of *Panax notoginseng*, Kunming University of Science and Technology, Kunming 650500, China; 4Institute of Biology Leiden, Leiden University, 2333BE Leiden, The Netherlands

**Keywords:** *Gastrodia elata* tuber, quality evaluation, HPLC, QAMS

## Abstract

*Gastrodia elata* (*G. elata*) tuber is a valuable herbal medicine used to treat many diseases. The procedure of establishing a reasonable and feasible quality assessment method for *G. elata* tuber is important to ensure its clinical safety and efficacy. In this research, an effective and comprehensive evaluation method for assessing the quality of *G. elata* has been developed, based on the analysis of high performance liquid chromatography (HPLC) fingerprint, combined with the quantitative analysis of multi-components by single marker (QAMS) method. The contents of the seven components, including gastrodin, *p*-hydroxybenzyl alcohol, *p*-hydroxy benzaldehyde, parishin A, parishin B, parishin C, and parishin E were determined, simultaneously, using gastrodin as the reference standard. The results demonstrated that there was no significant difference between the QAMS method and the traditional external standard method (ESM) (*p* > 0.05, RSD < 4.79%), suggesting that QAMS was a reliable and convenient method for the content determination of multiple components, especially when there is a shortage of reference substances. In conclusion, this strategy could be beneficial for simplifying the processes in the quality control of *G. elata* tuber and giving references to promote the quality standards of herbal medicines.

## 1. Introduction

*Gastrodia elata (G. elata)* Blume is a traditional medicinal herb that has been used in oriental countries, for centuries, to treat general paralysis, headaches, dizziness, rheumatism, convulsion, and epilepsy [1,2]. Modern pharmacological studies have demonstrated that the extracts of *G. elata* tuber and some compounds that originate from it, possesses wide-reaching biological activities, including anti-tumor, anti-virus, memory-improving, anti-oxidation, and anti-aging actions [3,4,5]. Nowadays, it is also widely used as a sub-material in food and Chinese Patent Medicines (CPM) [6], and this herbal medicine is also listed as one of the functional foods approved by the Ministry of Health in China [7,8]. As the wild *G. elata* is not sufficient enough for commercial large-scale exploitation, its artificial cultivation in medicine has become essential, to meet the increasing requirement of markers [6]. Due to their high medicinal value, *G. elata* tubers have been cultivated and produced in many areas of Asia, like China and Korea, which could lead to great differences in quality and, possibly, could lead to differences in the following clinical efficacies. Many studies have indicated that the efficacy and quality of herbal medicines are somewhat different depending on the cultivation soil and climate, based on the geographic origin, even when coming from the same species [9,10]. Therefore, a reasonable and effective method for the quality evaluation of *G. elata* tuber, plays an important role in its medication safety.

Gastrodin and its aglycone (*p*-hydroxybenzyl alcohol) are major components of the *G. elata* tuber, which are also markers for the quality control of this herbal medicine [11]. However, over 81 compounds from *G. elata* tuber have been currently isolated and identified. Along with the above two marker components, others like *p*-hydroxy benzaldehyde, parishin A, parishin B, parishin C, parishin E, and so on have also been reported to be correlated with the bioeffects of the *G. elata* tuber [12,13]. Accordingly, a qualitative analysis and quantification of one or two compounds, could be insufficient for a complete profile of the chemical characterization of the *G. elata* tuber, due to its complex compositions. In recent years, the chromatographic fingerprint analysis has been accepted as a strategy for the quality assessment of herbal medicines and preparations by the US Food and Drug Administration [14], State Food and Drug Administration of China [15], and the European Medicines Agency [16]. Since the fingerprint is characterized by more chemical information, the method is often used for the origin identification, species authentication, and quality control for herbal medicines, by observing the presence or absence of a limited number of peaks in the chromatographic fingerprints [17,18]. Therefore, the fingerprint analysis of high performance liquid chromatography (HPLC) was developed for the qualitative analysis of *G. elata* tuber.

A single standard to determine multiple components, also known as the quantitative analysis of multi-components by single marker (QAMS) [19], is a novel method designed for the quality evaluation of herbal medicines and related products [20]. Researchers have used QAMS to determine three components in Fructus Evodiae, simultaneously, by using rutaecarpine as the internal reference compound to calculate the relative correction factor of evodin and evodiamine [21]. To make up for the limitations of the fingerprint which cannot be quantified accurately, a QAMS method using berberine as the standard, was developed and validated for a simultaneous quantitative analysis of fourteen components [22]. This strategy could not only reduce the cost of the experiment and time of detection but could also be independent of the availability of all target ingredients [19]. Thus, the QAMS method was applied for a quantitative analysis of *G. elata* tuber.

This study aimed to establish a reliable and practical method, realizing both qualitative and quantitative analyses for *G. elata* tuber, via HPLC fingerprinting, combined with QAMS. The differences and similarities of the HPLC fingerprints were visually compared, using a hierarchical cluster analysis (HCA) and similarity analysis. The contents of seven major active constituents were accurately determined by both the QAMS method and external standard method (ESM), through which we hoped to offer a suitable and efficient approach for assessing the quality of *G. elata* tuber.

## 2. Results and Discussion

### 2.1. Optimization of the Chromatographic Conditions

As the components of *G. elata* tuber are very intricate, it is critical to optimize the chromatographic conditions, including favorable mobile phase systems, gradient elution systems, and the detection wavelength, to obtain an efficient separation of the target components. Lei [23] indicated that the HPLC fingerprints of *G. elata* tubers were the most informative, while the UV wavelength was 220 nm from HPLC-DAD-3D spectrum of *G. elata* tuber. So in this case, we chose the UV wavelength of 220 nm, to determinate the selected components. We chose acetonitrile-water containing 0.1% phosphoric acid system. The samples were dissolved in 60% methanol and ultrasound, for 60 min. We optimized the gradient elution system as Section 3.5, and 35 °C was selected as the proper temperature for analysis, while the flow rate was set at 1.0 mL/min. The S1 sample of *G. elata* tuber and the mixed standards containing seven reference substances were analyzed to obtain the HPLC fingerprints (Figure 1) under the conditions of Section 3.5, producing sharp and symmetrical chromatographic peak shapes, good separation, and preventing the peak tailing.

According to the retention time of each peak in the chromatogram [24], the peaks of 1, 2, 3, 4, 5, 6, and 7 were identified to be gastrodin, *p*-hydroxybenzyl alcohol, parishin E, *p*-hydroxy benzaldehyde, parishin B, parishin C, and parishin A. The separation degree of each peak was greater than 1.5, in the present HPLC system, indicating the peaks were well-separated, under the chromatographic conditions.

### 2.2. Method Validation

#### 2.2.1. Linearity

The mixed reference solution containing all the reference substances was diluted in series, with 60% methanol, to obtain six different concentrations for the seven reference curves. The linearity of each analyte was assessed by plotting its calibration curve with different concentrations and the corresponding peak areas. The results were shown in Table 1. The high correlation coefficient values indicated that there was a good correlation between the concentration and peak area of the seven compounds, at a relatively wide range of concentrations. The correlation coefficient of more than 0.9990, indicated a satisfactory linearity. The calibration curve could be utilized for the quantitative analysis in the given concentration range. The standard solution of the individual analyte was diluted gradually, to determine its Limit of Detection (LOD) and Limit of Quantity (LOQ) with signal-to-noise ratio of 3:1 and 10:1, respectively. LOD and LOQ values for the analytes are also listed in Table 1.

#### 2.2.2. Precision, Stability, Repeatability, and Accuracy

The precision was evaluated according to the assay of S1, in which the solution was analyzed for six times in a day, to evaluate the intra-day precision, and was analyzed on three consecutive days, to evaluate the inter-day precision. Calculating the RSDs of each chromatographic peak, the results showed that the RSDs of gastrodin, *p*-hydroxybenzyl alcohol, parishin E, *p*-hydroxy benzaldehyde, parishin B, parishin C, and parishin A were 1.93%, 1.10%, 1.29%, 2.30%, 2.03%, 2.63%, and 0.89% (n = 6), respectively, indicating that the precision of the method was good.

The stability was tested with the S1 solution that was stored at room temperature (25 ± 5 °C) and analyzed at 0, 2, 4, 6, 8, 12, and 24 h, to calculate the RSDs. The results showed that the RSDs of gastrodin, *p*-hydroxybenzyl alcohol, parishin E, *p*-hydroxy benzaldehyde, parishin B, parishin C, and parishin A were 1.15%, 2.04%, 1.51%, 2.37%, 2.10%, 1.12%, and 2.25%, respectively, suggesting that the method was stable within 24 h.

In the repeatability test, six duplicates of S1 were extracted and analyzed, according to the sample preparation procedure, and the HPLC method. The RSDs of the peak areas were calculated. The results showed that the RSDs of gastrodin, *p*-hydroxybenzyl alcohol, parishin E, *p*-hydroxy benzaldehyde, parishin B, parishin C, and parishin A were 1.25%, 2.15%, 1.60%, 1.81%, 1.72%, 1.84%, and 1.60% (n = 6), respectively, indicating that the repeatability of the method was good.

In the accuracy test, certain amounts of the seven analytes’ standards were added to the *G. elata* tuber samples (S1), with the six replicates. Then, these seven mixed samples were treated, as in the method described above. Recovery rate was used as the evaluation index and calculated as Recovery rate (%) = (Found amount − Known amount) × 100%/Added amount. The RSD of the accuracy values of the seven components are shown in Table 2, respectively.

The HPLC method was validated in terms of precision, repeatability, stability, and accuracy, as shown in Table 2. The RSD of the precision values of the seven components were less than 2.63%. RSD values for the stability and the repeatability were less than 2.37% and 2.15%, respectively. The recovery rates of the analytes ranged from 91.80% to 98.05%, with the RSD values being lower than 2.90%. All results indicated that the developed method was stable, accurate, and repeatable. This established HPLC method could be applied for a simultaneous determination of gastrodin, *p*-hydroxybenzyl alcohol, parishin E, *p*-hydroxy benzaldehyde, parishin B, parishin C, and parishin A, in the *G. elata* tuber samples.

### 2.3. HPLC Fingerprints Analysis

The 21 batches of *G. elata* tuber samples from the different producing areas were prepared according to Section 3.3, and 10 μL of S1 sample solution was injected into the HPLC system according to the chromatographic conditions in Section 3.5, to obtain the fingerprints. The retention time was the horizontal axis and the peak area was the vertical axis; the 3D fingerprints of the 21 batches of *G. elata* tuber samples were established by the software Origin 9.0, as shown in Figure 2.

According to Figure 2, the seven peaks with stable and better shape were determined to be the major ones for the HPLC fingerprints of *G. elata* tubers. The peak areas of the seven peaks are shown in Table 3. The variance coefficients of the peak area were greater than 32.2 percent, indicating that the content of each marker component varied greatly from place to place. 

### 2.4. Similarity Analysis

According to the data of HPLC fingerprints in Figure 2, the similarity of HPLC fingerprints from the different producing regions were evaluated using the Similarity Evaluation System for chromatographic fingerprint of traditional Chinese medicines (TCM) (Version 2012), with correlation coefficient (median) on behalf of the similarity of HPLC fingerprints. We utilized the average correlation coefficient method of 21 batches of the samples for the multipoint correction, and the time window width was set to 0.5 [25], while the establishment of a common model was to generate a control fingerprints of the *G. elata* tuber. Compared with the reference fingerprint chromatogram (R), the similarities of the 21 batches of samples were higher than 0.96, indicating that the batch-to-batch consistency was good. The results suggested that those samples of *G. elata* tuber had a similar chemical composition, and the samples were collected from the same genus, even though they were from different producing countries or were produced under different processing conditions (Table 4). Therefore, the developed fingerprint by HPLC could be used as a practical tool for the qualitative identification of the *G. elata* tuber. 

### 2.5. Hierarchical Cluster Analysis (HCA)

Using the peak areas of the seven compounds from the 21 *G. elata* tuber samples as the clustering variable, the HCA of the standardized data was performed with the heat map software of Heml 1.0. The graph in Figure 3 illustrated that the samples could be categorized into three groups. Group 1 contained S1 and S2 from Zhaotong, Yunnan in China; Group 2 contained S19 and S20 tubers from South Korea; and Group 3 contained the rest of samples. From the result, the samples from the same producing area were not always classified into the same group. For example, Zhaotong has been considered as the Daodi production area (area which produces authentic and superior medicinal materials) of the *G. elata* tuber in China. However, samples 1 to 6 from Zhaotong, showed different levels and ratios of chemical components, which could be due to the variations in harvesting time, planting patterns, dying methods, and other factors. Additionally, the preliminary processing method also contributes to the differences in the chemical composition. For instance, *G. elata* tubers and slices from South Korea were classified into different categories. Therefore, it is insufficient to determine the quality of the *G. elata* tubers by only their producing areas or any other single factor. Although the HCA could be used to classify the *G. elata* tubers on the basis of the peak areas of the seven components, it was hard to tell which group had a better quality. Therefore, other methods for the quantitative analysis of *G. elata* tubers should be developed, to reflect the quality difference.

### 2.6. Quantitative Analysis of Multiple Components by Single Marker

Theoretically, the quantity (mass or concentration) of an analyte is in direct proportion of the detector response. Then, in multi-component quantitation, a typical botanical compound (readily available) might be selected as an internal standard and the relative correction factor (RCF) of this marker, and the other components can be calculated.

#### 2.6.1. Calculation of RCFs

It is of vital importance to select a proper internal referring standard for the accurate assay of multiple components in TCM. The component chosen as the internal referring substance should be stable, easily obtainable, and have relatively clear pharmacologic effects related to the clinical efficacy of the herbal medicine [26]. In this work, the gastrodin was used as an internal referring substance for its easy availability, lower cost, moderate retention value, and good stability.

In order to simultaneously determine the contents of the seven components in the *G. elata* tuber, by using the QAMS method, the relative correction factors (RCFs, *f_x_*) were first determined, according to the ratio of the peak areas and the ratio of the concentration between the gastrodin and other compounds, as described in Section 3.6. We calculated the RCFs of six components (shown in Table 5). 

#### 2.6.2. Results from the QAMS Method 

After preparing the sample solutions of *G. elata* tubers, they were injected into the HPLC system to obtain the peak areas. The contents of seven compounds were calculated, according to the calibration curves. Those scattered in the vicinity of the lowest concentration point on the standard curve were determined with a one point ESM. Meanwhile, the contents of the seven components of the *G. elata* tuber calculated according to QAMS method, are shown in Table 6. 

The validated traditional ESM and QAMS method were employed to test the 21 batches of *G. elata* tuber samples from the different producing areas, which were based on the principle of the linear relationship between a detector response and the levels of components within certain concentration ranges. The validation of the QAMS method might be implemented, based on *t*-test, correlation coefficient [27], RSD [28], and relative error [29], through a comparison with an external standard. Correlation coefficient, as a statistical parameter, ranging from 0 (no correlation) to 1 (complete correlation), reflecting the closeness of two variables, is often used in similarity assessments of traditional Chinese medicine fingerprints [30]. As shown in Table 7, Correlation coefficients of the assay results obtained from the two methods were calculated here; all coefficients were found to be >0.998. The data showed that the results of the two methods were highly correlated. Then, a *t*-test was performed for the calculated results, by the QAMS method, and the on detected results, by an external standard method. *p*-values of gastrodin, *p*-hydroxy benzyl alcohol, parishin E, *p*-hydroxy benzaldehyde, parishin B, parishin C and parishin A, were all >0.05. The relative error and RSD values were all lower than 5%. Above all, the results indicated that there was no significant difference between the data from the QAMS and the ESM method, indicating that the present QAMS method was reliable for the simultaneous quantification of the seven components of the *G. elata* tuber.

The results from the QAMS determination of the 21 batches of *G. elata* tuber samples showed the mean contents of 3.5275 mg·g^-1^, 0.9060 mg·g^−1^, and 0.3398 mg·g^−1^ for gastrodin, *p*-hydroxy benzyl alcohol, and *p*-hydroxy benzaldehyde; and 3.6511 mg·g^−1^, 9.5303 mg·g^−1^, 2.7901 mg·g^−1^, and 0.1766 mg·g^−1^ for the parishin E, parishin A, parishin B, and parishin C, respectively (Table 4). It was obvious that parishin A is one of the most abundant components in *G. elata* tuber, thus, is well-deserved as a reference substance and index for quality assessment and control of the *G. elata* tuber. Obvious inter-batch content variations could be found for all these components with the mean ranging from 0.1766 mg·g^−1^ to 9.5303 mg·g^−1^; these seven components in total averaged 20.7031 mg·g^−1^ in the *G. elata* tuber, for the 21 batches of samples. The data in Table 4 shows differences among various samples. To show the clear classification of the *G. elata* tuber samples, the QAMS method with chemometrics analysis was performed in the subsequent analyses.

Meanwhile, the results (Table 6) illustrated that there were remarkable differences in the contents of the seven components, in *G. elata* tubers from different regions, which could be attributed to the variations of genetics, plant origins, environmental factors, drying process, storage conditions, and so on. It was obvious that gastrodin is one of the most abundant components in *G. elata* tuber. Combined with its activities related to the efficacies of *G. elata* tuber [31], gastrodin is well-deserved as a reference substance and index for quality assessment and control of *G. elata* tuber.

In the Chinese Pharmacopoeia of 2015 edition, gastrodin and *p*-hydroxy benzyl alcohol are determined as the marker components for the quality control and evaluation of *G. elata* tuber. Despite their close correlation with the efficacies of *G. elata* tuber, gastrodin can transform to *p*-hydroxybenzyl alcohol, which is the aglycone and metabolite of gastrodin [32]. Fresh *G. elata* tubers have to be processed before being traded as materia medica in the market. During the steaming process, the change trend of the gastrodin content was often contrary to the one of *p*-hydroxybenzyl alcohol. When the content of gastrodin was increased, the content of *p*-hydroxybenzyl alcohol was generally decreased, and vice versa. Additionally, different processing methods will result in different variation of the contents of the two components. Choi et al. [33] applied drying methods of freeze drying, hot air, infrared ray, and steaming, to process *G. elata* tuber. The results showed that after steaming, the content of gastrodin in *G. elata* tuber processed by freeze drying was decreased, whereas, the content of *p*-hydroxybenzyl alcohol was increased. However, tubers processed by hot-air and infrared ray drying showed the opposite results. Such transformations between gastrodin and *p*-hydroxybenzyl alcohol might be due to the deglycosylation or glycosylation, during the processing. Since the herbal medicine in the global market is often processed or dried by different methods, which results in the fluctuation in the content of single component, it is relatively stable and more comprehensive to reflect on the quality of *G. elata* tuber by monitoring multiple components, instead of a single one.

## 3. Materials and Methods

### 3.1. Plant Material

Samples of *G. elata* tuber from different producing areas were collected, as shown in Table 8.

### 3.2. Chemicals

The reference standards of gastrodin (no. B21243, purity HPLC ≥ 98%), *p*-hydroxybenzyl alcohol (no. B20326, purity HPLC ≥ 98%), *p*-hydroxy benzaldehyde (no. B20327, purity HPLC ≥ 99%), parishin A (no. BP1063, purity HPLC ≥ 98%), parishin B (no. BP1064, purity HPLC ≥ 98%), parishin C (no. B20913, purity HPLC ≥ 98%), parishin E (no. BP1648, purity HPLC ≥ 98%) were purchased from Sichuan Victory Biological Technology Co., Ltd. (Sichuan, China), and their structures are shown in Figure 4 and Figure 5. Methyl alcohol was purchased from the Tianjin Fengchuan Chemical Reagent Technology Co. Ltd. Acetonitrile (HPLC grade) was purchased from Sigma-Aldrich, Inc. (St. Louis, MO, USA). Phosphoric acid was purchased from the Tianjin JinDongTianZheng Precision Chemical Reagent Factory. Ultrapure water was generated with an UPT-I-20T ultrapure water system (Yunnan Ultrapure Technology, Inc., Yunnan, China). All other chemicals used were of analytical grade.

### 3.3. Preparation of the Sample Solution

The 21 batches of dried *G. elata* tubers from different producing areas were crushed by a Wiggling high-speed Chinese medicine shredder, then powdered and sieved through a 40-mesh sieve. The sample solution of *G. elata* tuber was precisely absorbed (2.0 mg) and immersed in 25 mL volumetric flask, with 60% methanol. Additional 60% methanol was added to compensate for the weight loss after ultrasonic extraction for 60 min, and shaking it well. All solutions were filtered through 0.22 μm filter membranes, before being precisely injected into the HPLC system.

### 3.4. Reference Solution Preparation

The reference solution of *G. elata* tuber was prepared by accurately dissolving weighed samples of each compound in 60% methanol, making a mixture of 0.8 mg/mL of parishin A, 0.9 mg/mL of parishin B, 0.5 mg/mL of parishin E, 1.5 mg/mL of *p*-hydroxy benzaldehyde, 3.4 mg/mL of *p*-hydroxybenzyl alcohol, 0.9 mg/mL of gastrodin, 1.3 mg/mL of parishin C, mixed evenly. All the standard solutions were stored in a refrigerator at 4 °C, before use.

### 3.5. Chromatographic Procedures

The HPLC analysis of the *G. elata* tuber were done on an Agilent 1260 series system (Agilent Technologies, Santa Clara, CA, USA) consisting of a G1311B pump, a G4212B DAD detector, and a G1329B auto-sampler. The YMC-Tyiart C18 column (250 × 4.6 mm, 5 µm) was adopted for the analysis. The mobile phase consisted of A (0.1% phosphate solution) and B (acetonitrile). The gradient mode was as follows: 3–5% B for 0–11 min; 5% B for 11–18 min; 5–14% B for 18–31 min; 14% B for 31–38 min; 14–20% B for 38–48min; 20–24% B for 48–55 min; 24–80% B for 55–75 min; 80–100% B for 75–80 min; 100% B for 80–95 min; 100–70% B for 95–100 min; 70–50% B for 100–105 min; 50–30% B for 105–110 min; 30–3% B for 110–115 min; 3% B for 115–130 min. The flow rate was set at 1.0 mL/min. The detection wavelength was 220 nm. The column temperature was set at 35 °C and sample volume was 10 µL.

### 3.6. Theory of the QAMS Method

Methods for calculating the RCFs have been previously reported [24,37]. First, gastrodin was selected as the internal standard, and a multipoint method (Equation (1)) was used to calculate the relative correction factors (RCF) for *p*-hydroxy benzaldehyde, *p*-hydroxybenzyl alcohol, parishin A, parishin B, parishin E, and parishin C. Then the content of the measured component was calculated according to Equation (2) [38]. 

The RCFs were calculated using the calibration curves as follows:(1)$$f_{k/s}=\frac{a_{k}}{a_{s}}$$

The content of the measured component was calculated as follows:(2)$$C_{k}=\frac{A_{k}}{\left(A_{s}{\times f}_{k/s}\right)}$$
where, a_s_ is the ratio of the slope of internal standard reference calibration equations; a_k_ is the ratio of the slope of measured component calibration equations; A_k_ is the peak area of the measured component; and A_s_ is the peak area of the internal standard reference [37]. 

The content of the multi-marker components measured by QAMS was compared with results from ESM, to validate the methods of QAMS.

### 3.7. Data Analysis

We used the ESM and QAMS to calculate the seven components in 21 batches of *G. elata* tuber, to verify the feasibility of QAMS. At the same time, HCA was performed using the heat map software of Heml 1.0, to further investigate the difference among the *G. elata* tuber samples. The data were analyzed and evaluated by the Similarity Evaluation System for the chromatographic fingerprint of TCM (Version 2012), to evaluate similarities of the chromatographic profiles of the *G. elata* tuber.

## 4. Conclusions

In this study, the quality assessment method of *G. elata* tubers were established using QAMS methods, in combination with HPLC fingerprints analyses. The *G. elata* tubers from different areas were analyzed by HPLC fingerprints and the contents of the seven components in *G. elata* tuber samples was determined by the QAMS method. On the basis of these results, the quality of *G. elata* tubers could be quantified and better identified comprehensively by HCA of synthesis and similarity analysis. HPLC fingerprint analyses, combined with the QAMS methods, could be a powerful and reliable way to provide both qualitative insight and quantitative data for comprehensive quality assessment of the complex multi-component systems. QAMS combined with the HPLC fingerprint might offer a holistic phytochemical profile of botanicals, along with similarity analysis and HCA of synthesis, and the quality of *G. elata* tubers would be evaluated and better and more comprehensively identified. Moreover, in subsequent analyses, it is also necessary to combine the chemical analysis, biological evaluation, pharmacological activity, and other methods to evaluate the quality of *G. elata* tubers for better studying the clinical effect.

## Figures and Tables

**Figure 1 molecules-24-01521-f001:**
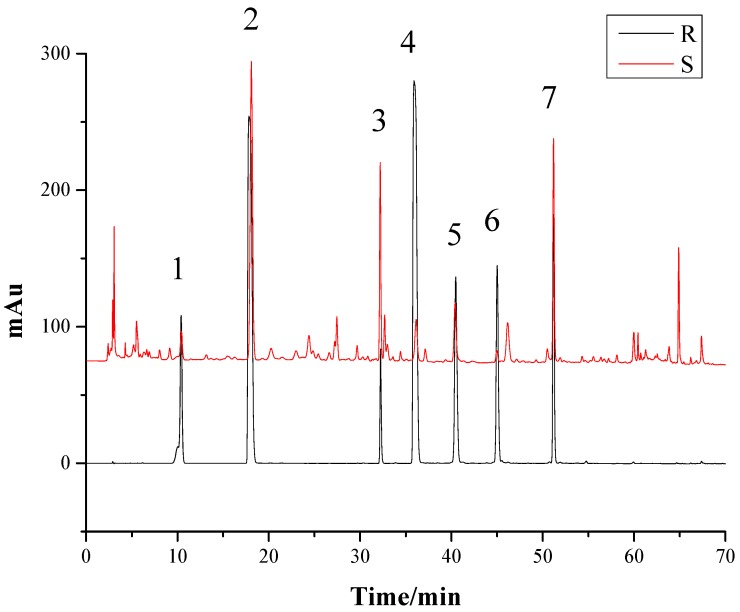
The HPLC fingerprints of the *Gastrodia elata* tuber sample and the mixed standards. R: The mixed standards; S: The *G. elata* tuber sample. 1—Gastrodin; 2—*p*-Hydroxy benzyl alcohol; 3—Parishin E; 4—*p*-Hydroxy benzaldehyde; 5—Parishin B; 6—Parishin C; 7—Parishin A.

**Figure 2 molecules-24-01521-f002:**
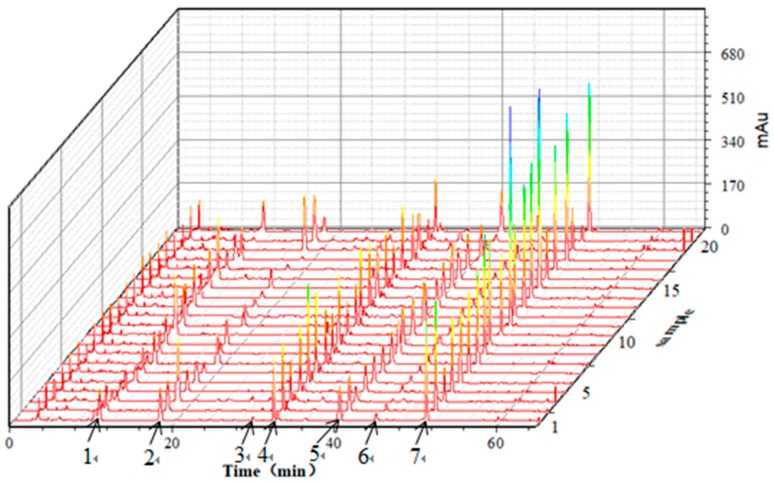
HPLC fingerprints of the 21 batches of *G. elata* tuber samples. 1—Gastrodin; 2—*p*-Hydroxy benzyl alcohol; 3—Parishin E; 4—*p*-Hydroxy benzaldehyde; 5—Parishin B; 6—Parishin C; 7—Parishin A.

**Figure 3 molecules-24-01521-f003:**
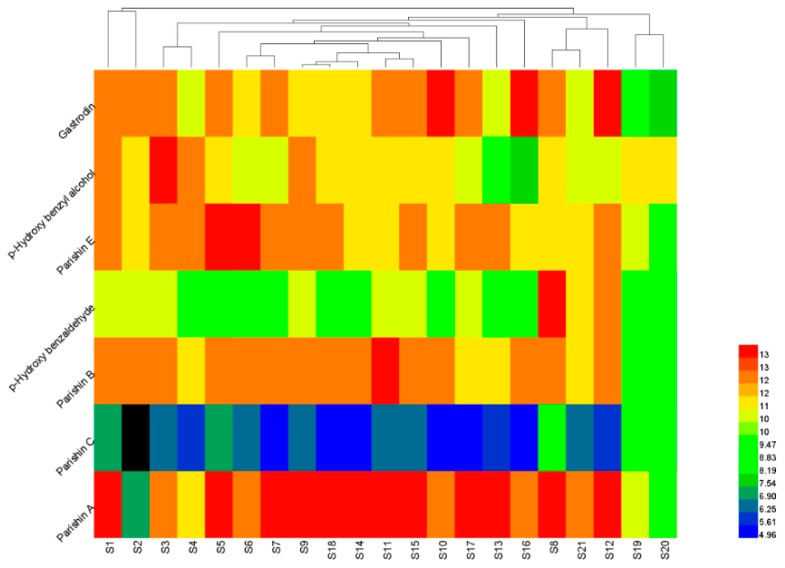
Clustering analysis graph of the 21 *G. elata* tuber samples.

**Figure 4 molecules-24-01521-f004:**
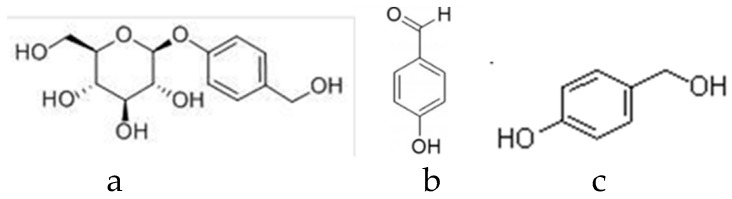
The structures of some compounds in the *G. elata* tuber. (**a**) Gastrodin [34], (**b**) *p*-hydroxy benzaldehyde [35], and (**c**) *p*-hydroxybenzyl alcohol [36].

**Figure 5 molecules-24-01521-f005:**
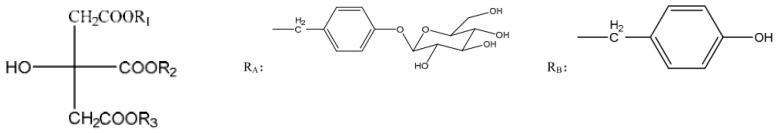
The structures of parishins in the *G. elata* tuber. The structure of parishins [13]: R_A_. parishin A, R_B_. parishin B, R_C_. parishin C, R_E_. parishin E.

**Table 1 molecules-24-01521-t001:** The regression equations, Limit of Detection (LODs) and Limit of Quantity (LOQs) of seven components.

Analytes	Regression Equations	Linear Ranges (mg/mL)	R^2^	LOD (mg/mL)	LOQ (mg/mL)
Gastrodin	*Y* = 18634*X* − 264.07	1.906~6.483	0.9997	0.042	0.139
*p*-Hydroxybenzyl alcohol	*Y* = 39300*X* + 42.955	0.075~1.773	0.9995	0.001	0.003
Parishin E	*Y* = 14141*X* + 142.93	2.273~7.052	0.9997	0.037	0.122
*p*-Hydroxy benzaldehyde	*Y* = 52536*X* + 7.9174	0.079~2.588	1.0000	0.001	0.005
Parishin B	*Y* = 20791*X* + 6.7746	1.450~5.190	1.0000	0.004	0.015
Parishin C	*Y* = 31240*X* − 335.24	0.286~0.356	0.9997	0.005	0.015
Parishin A	*Y* = 11769*X −* 100.83	0.181~19.301	0.9995	0.020	0.070

**Table 2 molecules-24-01521-t002:** RSD of precision, stability, repeatability and accuracy for determination of seven components.

Analyte	Precision	Stability	Repeatability	Accuracy	
RSD (%)	RSD (%)	RSD (%)	RSD (%)	Mean (%)	RSD (%)
Gastrodin	1.93	1.15	1.25	92.05%	2.02%
*p*-Hydroxybenzyl alcohol	1.10	2.04	2.15	95.78%	1.09%
Parishin E	1.29	1.51	1.60	98.05%	2.90%
*p*-Hydroxy benzaldehyde	2.30	2.37	1.81	92.44%	0.25%
Parishin B	2.03	2.10	1.72	93.33%	1.32%
Parishin C	2.63	1.12	1.84	92.91%	2.10%
Parishin A	0.89	2.25	1.60	91.80%	1.36%

**Table 3 molecules-24-01521-t003:** The information and peak areas of the seven characteristic peaks in HPLC fingerprints of *G. elata* tubers.

No.	Peak Area of Seven Characteristic Peaks						
	Gastrodin	*p*-Hydroxy Benzyl Alcohol	Parishin E	*p*-Hydroxy Benzaldehyde	Parishin B	Parishin C	Parishin A
**S1**	1797.1	2249.5	2337.8	217.4	2263.6	420.7	4340.3
**S2**	1470.2	2144.3	2523.4	227.6	2526.6	462.2	4561.9
**S3**	623.8	4536.4	1528.4	301.7	1017	280.3	1487.8
**S4**	1325.1	1516.1	1412.1	116.7	1906.9	402.8	3150
**S5**	1659.3	2123.3	1991.3	111.9	2141.7	383.4	4006.4
**S6**	1161	1463.7	3734.3	108.1	1867.1	390.2	3167.6
**S7**	1492.8	663.8	2991.6	85.8	1818.7	392.5	3473.5
**S8**	1470.9	823.8	1573.2	127.1	2231.8	546.3	5104.3
**S9**	1898	1876.6	2572.7	82	2629.1	595.9	4430.8
**S10**	3816.6	136.2	1316.2	110.1	2663.3	441.9	3383.6
**S11**	2353.9	970.8	1563.7	131.9	3073.4	789.5	9224.6
**S12**	1794	830	2577.2	45.7	2141.8	101.1	3845.8
**S13**	2344.5	572.4	2363.1	57.6	2039.1	499.2	5019.1
**S14**	1369.4	427.8	1961.6	41.9	2408.9	622.1	5512.4
**S15**	2177.5	1270.2	2076.6	56.6	3133.4	791.2	8184.9
**S16**	3322.1	108.1	1240.9	73.1	1935.1	357.8	2127.8
**S17**	1081.8	322.8	2365	104.7	2363.6	500.7	5062.6
**S18**	1893.7	270.9	1719.4	78.3	2475.6	823	6072.7
**S19**	380.4	4012.7	1414.3	617.3	781.5	136.4	1789.1
**S20**	300.9	3287.7	878.9	564	479.5	102.3	687.1
**S21**	2175.1	1076.8	2057.2	143.7	2826.4	94.7	6278.5
**C.V. (%) ^1^**	49.7	85.2	33.3	96.5	32.2	50.1	47.9

^1^ C.V. (%) = δ/µ × 100, δ—The standard deviation of peak area and µ—The average value of each peak area.

**Table 4 molecules-24-01521-t004:** Similarity of the *G. elata* tuber samples.

No.	Similarity	No.	Similarity	No.	Similarity	No.	Similarity
S1	0.983	S7	0.970	S13	0.988	S19	0.990
S2	0.987	S8	0.988	S14	0.982	S20	0.988
S3	0.983	S9	0.989	S15	0.979	S21	0.988
S4	0.975	S10	0.982	S16	0.989	*R*	1.000
S5	0.983	S11	0.987	S17	0.980		
S6	0.975	S12	0.990	S18	0.964		

**Table 5 molecules-24-01521-t005:** Relative correction factor (RCF) values of six components of the *G. elata* tuber.

Instrument	Chromatogram Column	RCF Values	
Agilent 1260	YMC-Tyiart C18 (250 × 4.6 mm, 5 μm)	***f_P_*_-hydroxy benzyl alcohol/gastrodin_**	2.1090
		***f*_parishin E/gastrodin_**	0.7589
		***f_P_*_-hydroxy benzaldehyde/gastrodin_**	2.8194
		***f*_parishin B/gastrodin_**	1.1156
		***f*_parishin C/gastrodin_**	1.6771
		***f*_parishin A/gastrodin_**	0.6316

**Table 6 molecules-24-01521-t006:** Contents of the seven components in *G. elata* tubes determined by the external standard method (ESM) and the quantitative analysis of multi-components by single marker (QAMS) methods (mg·g^-1^) ^1^.

No.	Gastrodin	*p*-Hydroxy Benzyl Alcohol		Parishin E		*p*-Hydroxy Benzaldehyde		Parishin B		Parishin C		Parishin A		Total
		ESM	QAMS	ESM	QAMS	ESM	QAMS	ESM	QAMS	ESM	QAMS	ESM	QAMS	
**S1**	5.23 ± 0.16	1.77 ± 0.05	1.82 ± 0.05	5.14 ± 0.01	5.35 ± 0.03	0.24 ± 0.01	0.25 ± 0.06	3.54 ± 0.14	3.60 ± 0.05	0.18 ± 0.00	0.19 ± 0.01	11.58 ± 0.45	11.71 ± 0.49	27.68
**S2**	4.51 ± 0.38	1.61 ± 0.08	1.64 ± 0.06	2.27 ± 0.14	2.38 ± 0.08	0.24 ± 0.01	0.25 ± 0.01	3.43 ± 0.21	3.47 ± 0.01	0.13 ± 0.00	0.14 ± 0.00	0.18 ± 0.00	0.00 ± 0.00	12.38
**S3**	2.44 ± 0.28	1.07 ± 0.08	1.09 ± 0.10	4.82 ± 0.25	5.00 ± 0.04	0.26 ± 0.01	0.26 ± 0.02	2.84 ± 0.10	2.88 ± 0.04	0.16 ± 0.02	0.16 ± 0.02	8.33 ± 0.31	8.32 ± 0.11	19.91
**S4**	1.35 ± 0.02	3.36 ± 0.12	3.39 ± 0.08	2.62 ± 0.06	2.62 ± 0.23	0.23 ± 0.01	0.24 ± 0.04	1.45 ± 0.04	1.46 ± 0.09	0.16 ± 0.00	0.16 ± 0.10	4.23 ± 0.12	4.10 ± 0.31	13.41
**S5**	3.41 ± 0.38	1.55 ± 0.10	1.58 ± 0.07	2.31 ± 0.22	2.36 ± 0.10	0.12 ± 0.02	0.13 ± 0.01	2.49 ± 0.12	2.51 ± 0.11	0.20 ± 0.00	0.20 ± 0.01	8.94 ± 0.63	8.92 ± 0.29	19.03
**S6**	1.91 ± 0.12	1.03 ± 0.07	1.06 ± 0.10	7.05 ± 0.13	7.26 ± 0.23	0.23 ± 0.02	0.24 ± 0.05	2.69 ± 0.03	2.73 ± 0.17	0.17 ± 0.00	0.17 ± 0.02	7.64 ± 0.31	7.63 ± 0.61	20.72
**S7**	3.12 ± 0.01	0.51 ± 0.00	0.531 ± 0.00	6.02 ± 0.14	6.20 ± 0.09	0.21 ± 0.01	0.21 ± 0.00	2.71 ± 0.02	2.73 ± 0.01	0.15 ± 0.00	0.16 ± 0.00	8.87 ± 0.02	8.86 ± 0.08	21.58
**S8**	3.06 ± 0.10	0.62 ± 0.01	0.64 ± 0.01	3.00 ± 0.18	3.13 ± 0.17	2.59 ± 0.02	2.60 ± 0.05	3.25 ± 0.15	3.27 ± 0.01	0.15 ± 0.01	0.14 ± 0.04	12.77 ± 0.58	12.75 ± 0.54	25.44
**S9**	2.85 ± 0.37	1.22 ± 0.18	1.24 ± 0.20	4.54 ± 0.03	4.69 ± 0.06	0.28 ± 0.01	0.28 ± 0.01	3.61 ± 0.06	3.63 ± 0.15	0.17 ± 0.01	0.16 ± 0.05	10.78 ± 0.15	10.77 ± 0.08	23.44
**S10**	5.89 ± 0.22	0.10 ± 0.01	0.10 ± 0.01	3.37 ± 0.24	3.52 ± 0.23	0.11 ± 0.01	0.11 ± 0.06	3.84 ± 0.13	3.91 ± 0.05	0.15 ± 0.02	0.16 ± 0.00	7.90 ± 0.67	7.91 ± 0.63	21.36
**S11**	4.74 ± 0.37	0.69 ± 0.08	0.71 ± 0.08	3.40 ± 0.22	3.55 ± 0.22	0.26 ± 0.02	0.26 ± 0.04	5.191 ± 0.09	5.23 ± 0.02	0.18 ± 0.01	0.19 ± 0.00	26.70 ± 0.46	26.93 ± 0.54	41.15
**S12**	7.10 ± 0.27	0.65 ± 0.04	0.66 ± 0.01	4.88 ± 0.23	5.04 ± 0.11	0.08 ± 0.12	0.08 ± 0.02	3.03 ± 0.16	3.18 ± 0.01	0.15 ± 0.02	0.15 ± 0.00	9.43 ± 0.54	9.80 ± 0.10	25.32
**S13**	4.03 ± 0.03	0.37 ± 0.01	0.39 ± 0.01	3.69 ± 0.11	3.86 ± 0.11	0.16 ± 0.01	0.17 ± 0.02	2.37 ± 0.06	2.41 ± 0.06	0.15 ± 0.00	0.15 ± 0.00	10.27 ± 0.24	10.33 ± 0.24	21.04
**S14**	2.59 ± 0.03	0.19 ± 0.16	0.20 ± 0.00	2.97 ± 0.08	3.10 ± 0.07	0.20 ± 0.00	0.21 ± 0.02	2.88 ± 0.06	2.90 ± 0.05	0.15 ± 0.01	0.15 ± 0.00	11.61 ± 0.37	11.58 ± 0.33	20.59
**S15**	4.29 ± 0.15	0.92 ± 0.04	0.94 ± 0.04	3.76 ± 0.16	3.90 ± 0.16	0.31 ± 0.02	0.31 ± 0.07	4.27 ± 0.17	4.30 ± 0.10	0.13 ± 0.00	0.14 ± 0.00	19.30 ± 0.83	19.42 ± 0.84	32.98
**S16**	6.48 ± 0.21	0.08 ± 0.00	0.08 ± 0.00	2.35 ± 0.11	2.41 ± 0.13	0.11 ± 0.00	0.11 ± 0.05	2.75 ± 0.11	2.78 ± 0.04	0.15 ± 0.01	0.15 ± 0.01	5.24 ± 0.15	5.16 ± 0.20	17.15
**S17**	5.09 ± 0.39	0.13 ± 0.22	0.13 ± 0.26	4.20 ± 0.05	4.39 ± 0.05	0.24 ± 0.00	0.25 ± 0.07	1.54 ± 0.17	1.56 ± 0.08	0.15 ± 0.01	0.14 ± 0.01	11.65 ± 0.22	11.77 ± 0.23	23.01
**S18**	3.71 ± 0.05	0.20 ± 0.01	0.20 ± 0.01	4.22 ± 0.09	4.41 ± 0.13	0.20 ± 0.00	0.20 ± 0.04	3.47 ± 0.08	3.52 ± 0.03	0.15 ± 0.00	0.16 ± 0.01	14.60 ± 0.21	14.76 ± 0.24	26.54
**S19**	0.42 ± 0.01	1.23 ± 0.02	1.26 ± 0.02	1.10 ± 0.01	1.14 ± 0.01	0.18 ± 0.01	0.18 ± 0.00	0.46 ± 0.01	0.47 ± 0.01	0.38 ± 0.01	0.39 ± 0.04	1.99 ± 0.06	1.98 ± 0.05	5.76
**S20**	0.36 ± 0.02	1.01 ± 0.01	1.04 ± 0.01	0.64 ± 0.00	0.65 ± 0.00	0.12 ± 0.00	0.11 ± 0.00	0.28 ± 0.00	0.29 ± 0.00	0.30 ± 0.01	0.30 ± 0.03	0.83 ± 0.09	0.82 ± 0.00	3.52
**S21**	1.50 ± 0.14	0.33 ± 0.00	0.34 ± 0.00	1.70 ± 0.40	1.71 ± 0.04	0.68 ± 0.02	0.69 ± 0.01	1.75 ± 0.02	1.78 ± 0.01	0.17 ± 0.00	0.16 ± 0.01	6.62 ± 0.03	6.61 ± 0.03	12.75
**Mean**	3.53		0.91		3.65		0.34		2.79		0.18		9.53	20.70

^1^ ESM—external standard method, and its content was determined by the calibration equation method; QAMS—quantitative analysis multi-components by single marker, and its content was determined by RCFs; RSD—relative standard deviation; Total—the sum of the six alkaloid contents in each batch.

**Table 7 molecules-24-01521-t007:** The relative error, RSD, correlation coefficient, and *p* values of the contents from the ESM and the QAMS ^1^.

No.	*p*-Hydroxy Benzyl Alcohol		Parishin E		*p*-Hydroxy Benzaldehyde		Parishin B		Parishin C		Parishin A	
	Relative Error	RSD	Relative Error	RSD	Relative Error	RSD	Relative Error	RSD	Relative Error	RSD	Relative Error	RSD
**S1**	2.38%	1.70%	4.04%	2.92%	2.39%	1.71%	1.76%	1.25%	1.47%	1.05%	1.15%	0.82%
**S2**	1.78%	1.27%	4.56%	3.30%	1.72%	1.23%	1.10%	0.78%	1.55%	1.11%	0.00%	0.00%
**S3**	2.38%	1.70%	3.72%	2.68%	1.94%	1.39%	1.37%	0.97%	1.56%	1.11%	0.02%	0.10%
**S4**	0.72%	0.51%	0.10%	0.07%	1.11%	0.79%	0.60%	0.43%	0.94%	0.67%	0.18%	2.13%
**S5**	1.52%	1.08%	2.13%	1.52%	2.14%	1.53%	0.90%	0.64%	1.11%	0.79%	0.03%	0.18%
**S6**	2.50%	1.79%	2.95%	2.12%	2.01%	1.44%	1.38%	0.98%	0.66%	0.47%	0.01%	0.07%
**S7**	3.39%	2.43%	2.84%	2.03%	1.70%	1.21%	0.97%	0.69%	3.09%	2.22%	0.02%	0.12%
**S8**	2.38%	1.70%	4.27%	3.08%	0.32%	0.23%	0.37%	0.26%	3.81%	2.74%	0.02%	0.09%
**S9**	1.65%	1.17%	3.18%	2.29%	1.17%	0.83%	0.60%	0.43%	0.74%	0.52%	0.02%	0.10%
**S10**	1.29%	0.92%	4.24%	3.06%	2.81%	2.01%	1.59%	1.13%	3.05%	2.19%	0.21%	0.15%
**S11**	2.57%	1.84%	4.20%	3.03%	1.38%	0.98%	0.74%	0.53%	4.84%	3.51%	0.88%	0.62%
**S12**	0.77%	0.54%	3.22%	2.32%	3.59%	2.59%	4.72%	3.42%	0.31%	0.22%	3.81%	2.75%
**S13**	4.77%	3.45%	4.54%	3.29%	2.41%	1.73%	1.48%	1.05%	0.90%	0.64%	0.61%	0.43%
**S14**	4.79%	3.47%	4.31%	3.12%	1.15%	0.82%	0.39%	0.27%	0.66%	0.47%	0.27%	0.19%
**S15**	1.95%	1.39%	3.72%	2.68%	1.10%	0.78%	0.59%	0.42%	4.61%	3.34%	0.61%	0.44%
**S16**	4.71%	3.41%	2.10%	1.50%	2.69%	1.93%	1.15%	0.82%	0.39%	0.28%	1.44%	1.02%
**S17**	0.76%	0.54%	4.31%	3.12%	2.23%	1.60%	1.61%	1.15%	3.77%	2.72%	0.99%	0.70%
**S18**	0.35%	0.25%	4.31%	3.11%	2.23%	1.59%	1.43%	1.02%	4.95%	3.59%	1.14%	0.81%
**S19**	2.54%	1.82%	3.76%	2.71%	2.50%	1.79%	2.33%	1.67%	0.94%	0.67%	0.38%	0.27%
**S20**	3.25%	2.33%	1.43%	1.02%	3.59%	2.59%	3.34%	2.40%	0.07%	0.05%	1.13%	0.80%
**S21**	2.53%	1.81%	0.70%	0.50%	1.75%	1.24%	1.70%	1.21%	2.10%	1.50%	0.20%	0.14%
**Correlation coefficient**	0.999 **		0.999 **		0.999 **		1.000 **		0.998 **		0.999 **	
***p* values**	0.940		0.802		0.978		0.923		0.960		0.986	

^1^ RSD—relative standard deviation; *p* values—the paired *t*-test results; ESM—external standard method, and its content was determined by the calibration equation method; QAMS—quantitative analysis multi-components by single marker, and its content was determined by RCFs; ** *p* < 0.01.

**Table 8 molecules-24-01521-t008:** The information of *G. elata* tubers from different producing areas.

No.	Sample	Producing Areas	No.	Sample	Producing Areas
**S1**	*G. elata* tubers	Zhaotong, Yunnan, China	**S12**	*G. elata* tubers	Enshi, Hubei, China
**S2**	*G. elata* tubers	Zhaotong, Yunnan, China	**S13**	*G. elata* tubers	Yichang, Hubei, China
**S3**	*G. elata* tubers	Zhaotong, Yunnan, China	**S14**	*G. elata* tubers	Hanzhong, Shanxi, China
**S4**	*G. elata* tubers	Zhaotong, Yunnan, China	**S15**	*G. elata* tubers	Qinling, Shanxi, China
**S5**	*G. elata* tubers	Zhaotong, Yunnan, China	**S16**	*G. elata* tubers	Qinchuan, Sichuang, China
**S6**	*G. elata* tubers	Zhaotong, Yunnan, China	**S17**	*G. elata* tubers	Longnan, Gansu, China
**S7**	*G. elata* tubers	Lijiang, Yunnan, China	**S18**	*G. elata* tubers	Anhui, China
**S8**	*G. elata* tubers	Bijie, Guizhou, China	**S19**	*G. elata* tubers	Moju, South Korea
**S9**	*G. elata* tubers	Zhengyuan, Guizhou, China	**S20**	*G. elata* tubers	Chun chuan, South Korea
**S10**	*G. elata* tubers	Qiandongnan, Guizhou, China	**S21**	*G. elata* tuber slices	Yingyang, South Korea
**S11**	*G. elata* tubers	Bijie, Guizhou, China

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
