# Peer review of "Quality Evaluation of Gastrodia Elata Tubers Based on HPLC Fingerprint Analyses and Quantitative Analysis of Multi-Components by Single Marker"

_molecules, 2019, doi:10.3390/molecules24081521_

Round 1

Reviewer 1 Report

The manuscript deals with quality evaluation of Gastrodia elata tubers based on HPLC fingerprint analyses and quantitative analysis of multi-components by single marker.

The paper is original and it can be consider as relevant in analytical chemistry methods.

 I consider that, after a thorough reading of the manuscript, this paper can be published in Molecules without change.

In the extension of the manuscript assessment, I would like to add that the work is well-thought-out and clearly written, the experiment is well planned and presented in the discussion. The assessment of the quality of plant extracts is a problem especially in the absence of commercially available standards. In this context, the manuscript is worth presenting wider community. Nevertheless, the evaluation method used by the authors is known from the literature. This manuscript is therefore another on the subject. Hence my assessment

Author Response

Dear reviewer

On behalf of my co-authors, we thank you very much for your reviewe. We would like to express our great appreciation to you.

Thank you and best regards.

Reviewer 2 Report

Authors reported the analytical method and evaluated the characteristics of G. elata.

I think manuscript was well described. And markers presented by author also well analysed.

This manuscript is acceptable to publish.

The authors described the simultanuous analysis of 7 active compounds  using HPLC. Seven active component is good markers for G. elata. they compared the two types of analytical methods, ESM and QAMS. Both ESM and QAMS are considered to be suitable for quantitative analysisof G.elata tuber. And authors analysed the various samples in China and Korea using their method. Thus, results of this study provide an excellent foundation for future development of G.elata tuber based medicinal preparations.

Author Response

(The authors gave the same response as above.)

Reviewer 3 Report

Authors present an interesting study entitled “Quality evaluation of Gastrodia elata tubers based on HPLC fingerprint analyses and quantitative analysis of multi-components by single marker” The article is well structured and written. However, the authors should improve some aspect.:

Pag 2 The author should improve bibliographical references :  

D’Archivio A.A.; Di Donato F.; Foschi M.; Maggi M.A.; Ruggieri F. Uhplc analysis of saffron (crocus sativus l.): Optimization of separation using chemometrics and detection of minor crocetin esters. Molecules 23, Issue 8, 2018, 1851

Hu, M., Yan, H., Fu, Y., Jiang, Y., Yao, W., Yu, S., Zhang, L., Wu, Q., Ding, A., Shan, M., Optimal extraction study of gastrodin-type components from gastrodia elata tubers by response surface design with integrated phytochemical and bioactivity evaluation Molecules 24, 2019, Article number 547

D'Archivio, A.A.; Giannitto, A.; Maggi, M.A.; Ruggieri, F. Geographical classification of Italian saffron (Crocus sativus L.) based on chemical constituents determined by high-performance liquid-chromatography and by using linear discriminant analysis  Food Chemistry, 212, 2016, Pages 110-116

Zuo, Y., Deng, X., Wu, Q., Discrimination of gastrodia elata from different geographical origin for quality evaluation using newly-build near infrared spectrum coupled with multivariate analysis Molecules 23, 2018, Article number 1088

Pag 3 the run time of chromatogram reported in fig 1 it is different from the one in fig 2, why?

Pag 9 :The author must correct the table 4, and report the data with a significant number of correct digits (for example gastrodin 5.23±0.16  instead 5.2326 ± 0.1624  , 4.5±0.4 instead 4.5062±0.3839…..).

Author Response

Dear reviewer

On behalf of my co-authors, we thank you very much for your comments. We would like to express our great appreciation to you.

We have made the following modifications to the article according to your comments

Point 1: Pag 2 The author should improve bibliographical references:
       D’Archivio A.A.; Di Donato F.; Foschi M.; Maggi M.A.; Ruggieri F. Uhplc analysis of saffron (crocus sativus l.): Optimization of separation using chemometrics and detection of minor crocetin esters. Molecules 23, Issue 8, 2018, 1851
       Hu, M., Yan, H., Fu, Y., Jiang, Y., Yao, W., Yu, S., Zhang, L., Wu, Q., Ding, A., Shan, M., Optimal extraction study of gastrodin-type components from gastrodia elata tubers by response surface design with integrated phytochemical and bioactivity evaluation Molecules 24, 2019, Article number 547
       D'Archivio, A.A.; Giannitto, A.; Maggi, M.A.; Ruggieri, F. Geographical classification of Italian saffron (Crocus sativus L.) based on chemical constituents determined by high-performance liquid-chromatography and by using linear discriminant analysis  Food Chemistry, 212, 2016, Pages 110-116
       Zuo, Y., Deng, X., Wu, Q., Discrimination of gastrodia elata from different geographical origin for quality evaluation using newly-build near infrared spectrum coupled with multivariate analysis Molecules 23, 2018, Article number 1088

Response 1: Thank you for this valuable feedback. We have been improved bibliographical references in pag 2 to describe the Introduction.

Point 2: Pag 3 the run time of chromatogram reported in fig 1 it is different from the one in fig 2, why?

Response 2: Thanks to your comment, the run time of chromatogram reported in figure 1 is same as the one in figure 2. Since it is not clear to see the run time of chromatogram in figure 2 visually, we marked the peak in the figure 2.

Point 3: Pag 9: The author must correct the table 4, and report the data with a significant number of correct digits (for example gastrodin 5.23±0.16 instead 5.2326 ± 0.1624, 4.5±0.4 instead 4.5062±0.3839…..).

Response 3: We are extremely grateful to you for pointing out this problem. We have revised Table 4 and adjusted the text where highlighted.

Reviewer 4 Report

The manuscript developed a HPLC method for the simultaneous quantification of 7 components in Gastrodia elata tubers using one marker. This work is very useful for further developing a standard for the quality control of Gastrodia elata tubers. However, the method validation of the developed analytical method need to be further investigated before publication on the journal of MOLECULES.

1. In this study, UV wavelength of 220 nm was used for sample determination. Is it suitable for all selected components? This should be further explained in the text.

2. The limit of detection (LOD) and limit of quantification (LOQ) for all components need to be provided.

3. The tests for reproducibility, repeatability, inter-/intra-day precision and accuracy of the analytical method need to be conducted.

4. The recovery of the method for sample preparation need to be further investigated.

5. The robustness and ruggedness of the analytical method should be determined and provided.

Author Response

Dear reviewer

On behalf of my co-authors, we thank you very much for your comments. We would like to express our great appreciation to you.

We have made the following modifications to the article according to your comments

Point 1: In this study, UV wavelength of 220 nm was used for sample determination. Is it suitable for all selected components? This should be further explained in the text.

Response 1: We are extremely grateful to you for pointing out the problem, we have added “Lei (Lei, Y., C. Authenticity identification and quality assessment of Gastrodia tuber (Tianma) based on chemical characteristics. Chendouo University of TCM, China, 2015, 5.) indicated that the HPLC fingerprints of G. elata tubers were the most informative while UV wavelength was 220 nm from HPLC-DAD-3D spectrum of G. elata tuber. So in this case, we chose UV wavelength of 220 nm to determinate selected components.” to explained in the text.

Point 2: The limit of detection (LOD) and limit of quantification (LOQ) for all components need to be provided.

Response 2: Thanks to your comment, we have added new data of the LOD and LOQ for all components in Table 1.

Point 3: The tests for reproducibility, repeatability, inter-/intra-day precision and accuracy of the analytical method need to be conducted.

Response 3: As suggested by you we have conducted for precision, repeatability and stability of the analytical method in “2.2.2. Precision, Stability, Repeatability and Accuracy” and added new table as Table 2, to clarify the point that the developed method is stable, accurate and repeatable.

Point 4: The recovery of the method for sample preparation need to be further investigated.

Response 4: Thank you for this valuable feedback. We have added new date of the recovery in “2.2.2. Precision, Stability, Repeatability and Accuracy”.

Point 5: The robustness and ruggedness of the analytical method should be determined and provided.

Response 5: Thank you for your suggestion. The suggested experiment is valid and would provide additional information about the analytical method. We are curious what the results would be. However, we didn't research that provide the answer now, basing on “robustness tests are not obligatory yet according to the ICH (International Conference on Harmonisation of Technical Requirements for the Registration of Pharmaceuticals for HumanUse) guidelines” (Dejaegher B, Heyden Y V. Ruggedness and robustness testing. Journal of Chromatography A, 2007, 1158(1):138-157.). We will follow your suggestion to determine and provide the robustness and ruggedness of the analytical method in the following research.

Thank you and best regards.

Round 2

Reviewer 4 Report

The manuscript has been well improved and is acceptable for publication on the journal of Molecules.